# Taking a critical stance towards mixed methods research: A cross-disciplinary qualitative secondary analysis of researchers' views

Sergi Fàbregues[1]*, Elsa Lucia Escalante-Barrios[2], José Francisco Molina-Azorin[3], Quan Nha Hong[4], Joan Miquel Verd[5]

1 Department of Psychology and Education, Universitat Oberta de Catalunya, Barcelona, Spain, 2 Department of Education, Universidad del Norte, Barranquilla, Colombia, 3 Department of Management, Universidad de Alicante, Alicante, Spain, 4 Evidence for Policy and Practice Information and Co-ordinating Centre (EPPI-Centre), University College London, London, United Kingdom, 5 Centre d'Estudis Sociologics sobre la Vida Quotidiana i el Treball (Sociological Research Centre on Everyday Life and Work—QUIT), Universitat Autonoma de Barcelona, Barcelona, Spain

* sfabreguesf@uoc.edu, sergi.fabregues@gmail.com

**Data Availability Statement:** Data cannot be shared publicly as participants did not give consent

## Abstract

Recent growth and institutionalization in the field of mixed methods research has provided fertile ground for a wide range of thoughtful criticism of how this research approach has been developed and conceptualized by some members of the mixed methods community. This criticism reflects the increasing maturity of the field as well as the different theoretical perspectives and methodological practices of researchers in different disciplines. While debates addressing these criticisms are likely to lead to valuable insights, no empirical studies have been carried out to date that have investigated researchers' critical views on the development and conceptualization of mixed methods research. This study examines the criticisms of the mixed methods field raised by a cross-national sample of researchers in education, nursing, psychology, and sociology. We carried out a secondary analysis of semi-structured interviews with 42 researchers and identified 11 different criticisms, which we classified in four domains: essence of mixed methods, philosophy, procedures, and politics. The criticisms related to the procedures domain were equally distributed among the four disciplines, while those related to the essence, philosophy and politics domains were more common among sociologists. Based on our findings, we argue that the divergence of views on foundational issues in this field reflects researchers' affiliation to different communities of practice, each having its own principles, values, and interests. We suggest that a greater awareness of this divergence of perspectives could help researchers establish effective collaboration and anticipate potential challenges when working with researchers having different methodological approaches.

for their transcripts to be shared in this manner. Since the consent statement approved by the Institutional Review Board (IRB) of the Universitat Autònoma de Barcelona, and signed by the participants, did not include the provision that data would be made publicly available, we do not have participant consent to share this data. Also, the content is sensitive, and participants could be identified. Requests for anonymized data can be made to the Principal Investigator of the study, Sergi Fàbregues (sfabreguesf@uoc.edu) or the Institutional Review Board (IRB) of the Universitat Autònoma de Barcelona (ceeah@uab.cat).

**Funding:** The authors received no specific funding for this work.

**Competing interests:** The authors have declared that no competing interests exist.

## Introduction

Since the 1990s, mixed methods research as a distinct methodology has seen vigorous institutionalization [1,2] with the launch of a specialized journal (*Journal of Mixed Methods Research*) in 2007, the establishment of the Mixed Methods International Research Association (MMIRA) in 2013, and the publication of a stream of handbooks, textbooks, and articles on the philosophy and practice of mixed methods. Mixed methods research is increasingly used in a wide range of disciplines, particularly in the social, educational, behavioral and health sciences [3–6]. Several scholars have argued that mixed methods inquiry can help researchers arrive at a more complete understanding of research problems, develop more robust quantitative instruments, and integrate several worldviews in a single research study [7,8].

A clear indication of the institutionalization of mixed methods research as a field is the expansion of the mixed methods community. Tashakkori, Johnson and Teddlie [9] have characterized this community as a group of scholars who share similar backgrounds, methodological orientations, philosophical assumptions, and views on research and practice. As distinct from researchers using only qualitative or quantitative approaches, these scholars often share similarities in training, research background, and professional affiliation. Those authors' view of the mixed methods community is consistent with Thomas Kuhn's preferred definition of paradigms, as cited by Morgan [10]. In disagreement with the view of paradigms as incompatible epistemological stances, Morgan [10], following Kuhn [11], advanced a more integrative notion of paradigms focused on shared beliefs and joint actions in a community of researchers. Denscombe [12] took this perspective one step further by proposing the notion of *communities of practice*, an idea originally developed by the educationalist Étienne Wenger. This notion brings us closer to a definition of paradigms as shared beliefs able to accommodate the diversity of perspectives and approaches that currently exists within the mixed methods community [12,13].

While the institutionalization of mixed methods as a field has helped to formalize and clarify research practices [14], this trend has also led to some criticism of the ways in which this approach has been developed and conceptualized by some members of the mixed methods community [15,16]. The critiques of the mixed methods field have already been summarized in two overviews published in the early 2010s. In the second edition of the *Handbook*, Tashakkori and Teddlie [17] outlined four frequent criticisms raised in the mixed methods literature, including overreliance on typologies and the higher status of quantitative versus qualitative research. One year later, Creswell [18] described some of the same criticisms in a summary of 11 key controversies. The emergence of this criticism testifies to the increasing maturity of the field and its progress towards what Creswell and Plano Clark [7] define as the period of reflection and refinement in mixed methods research. In their view, the mixed methods community should be honored that it has attracted critical attention and it should stimulate debate around the issues raised. Therefore, it is crucially important to address these criticisms in greater detail because such engagement will lead to valuable insights that could lay the basis for further discussion needed to ensure the healthy development of the field. Furthermore, addressing those criticisms is essential to enhance researchers' understanding of the complexity of the mixed methods field and to provide them with the awareness needed to deal with tensions that might emerge when working in teams with researchers subscribing to different methodological viewpoints [19,20].

Most of the criticisms of mixed methods so far have been formulated in the literature by several scholars specialized in theoretical aspects of methodology. However, it would be useful to find out whether other researchers share these criticisms and whether they may have formulated others. Scholars with an interest in mixed methods research come from different

academic disciplines that embody different theoretical and methodological perspectives. As Plano Clark and Ivankova [3] argue, these differences are highly likely to influence the ways in which those scholars view mixed methods as well as the questions they might raise regarding current ideas in the field. Therefore, it would be useful to examine the ways in which researchers' criticisms differ according to discipline.

The aim of the present study is to examine the criticisms of the mixed methods field raised by a cross-national sample of 42 researchers working in the disciplines of education, nursing, psychology, and sociology. We report a secondary analysis of semi-structured interviews originally conducted to describe how researchers operationalize and conceptualize the quality of mixed methods research [14]. The contribution of this article is twofold: (a) it is, to our knowledge, the first study based on an empirical approach to examining researchers' critical views on the development and conceptualization of mixed methods research, and (b) it enhances our understanding of the ways in which these critical views may be associated with different academic disciplines.

## Methods

This article reports a secondary analysis of data originally collected in a multiple-case study of the quality of mixed methods based on semi-structured interviews with researchers in the disciplines of education, nursing, psychology, and sociology. In line with Heaton's [21] definition of secondary analysis — which he calls *supplementary analysis* — as an "in-depth investigation of an emergent issue or aspect of the data which was not considered or fully addressed in the primary study" [21], in this article we re-analyze the original interview data in order to address the following two research questions (RQs): (RQ1) What criticisms of the mixed methods field are made by researchers in education, nursing, psychology, and sociology? and (RQ2) What differences and similarities can be identified in the criticisms reported by researchers working in different disciplines? In the following subsections, we provide a brief description of the sampling and data collection methods used in the original study, and of the procedures used in the secondary analysis of data. A more detailed explanation of procedures followed in the original study can be found in Fàbregues, Paré, and Meneses [22].

### The original study

**Sampling and recruitment of participants.** The disciplines of education, nursing, psychology, and sociology were selected for four main reasons: (1) professionals working in these disciplines contribute a relatively high proportion of mixed methods empirical articles and other methodological publications, (2) a considerable number of prevalence studies and methodological reviews on the use of mixed methods in subfields of these disciplines have been published, and (3) these disciplines are characterized by their clear disciplinary boundaries, and this characteristic offers the possibility of gaining useful comparative insights [6]. Criterion and maximum variation sampling were used to select the researchers who participated in the study [23]. In the criterion sampling, participants fulfilled two inclusion criteria: (1) they had carried out research primarily in one of the four disciplines mentioned above, and (2) they had contributed to at least one methodological publication on mixed methods research. Participant identification started with a systematic search for methodological publications on mixed methods published in English during or after 2003. Selected publications fulfilled the definitions of mixed methods suggested by either Creswell and Tashakkori [24] or Johnson, Onwuegbuzie, and Turner [25]. A number of characteristics of the first authors' profiles were extracted, including the field of expertise, the country of affiliation, and the job title. To ensure heterogeneity of the sample, maximum variation sampling was applied to authors meeting the

two inclusion criteria. An iterative approach was used to recruit 11 participants for each discipline. Sample size was based on recommendations found in the literature [26]. Potential participants were contacted using a prioritized list until a total of 44 participants had been recruited.

**Data collection.** Data collection involved semi-structured interviews. Questions from the interview guide were focused on the following topics: (a) participants' research background and methodological expertise, (b) participants' conceptualization of mixed methods research, and (c) how participants perceived the quality of the mixed methods approach in practice. Interviews were conducted using Skype, telephone and, in two cases, e-mail correspondence. Interviews were audio-recorded and transcribed, and average interview length was 49 minutes. We carried out a member-checking process by sending back to participants the transcriptions and summaries of key points of the interviews to confirm that the data accurately represented their views. At this stage, we also gave participants an opportunity to clarify or expand the statements they made during the interview.

**Trustworthiness.** Four strategies were used to enhance the trustworthiness of the original study. First, as explained above, all participants member-checked their transcribed interviews and summaries to confirm accuracy. Second, peer-debriefing was carried out during data collection by one researcher working together with another researcher familiar with mixed methods research who was not included in the sample. Third, an audit trail was used to record the decisions made during the study and to help researchers to reflect on the influence on the study findings of their own assumptions and disciplines. Fourth, the decisions taken during the analysis and interpretation of the data, as well as the disagreements arising during this stage, were discussed by the researchers until a consensus was reached.

## The secondary data analysis

While the original study aimed to examine researchers' views on the conceptualization and operationalization of the quality of mixed methods research, the aim of this secondary analysis of the same data is to examine researchers' critical views of commonly accepted concepts and practices in the mixed methods field. Ethical approval for secondary data analysis was included in the ethics application for the original study, which was approved by the Institutional Review Board of the Universitat Autònoma de Barcelona. Participants signed an informed consent before the interview. Of the 44 researchers who participated in the original study, two did not consent to the subsequent use of their interview data for a secondary analysis. Therefore, the information provided by these two researchers was not used in the present study. Table 1 shows the characteristics of the 42 participants finally included in this secondary analysis.

Qualitative content analysis as described by Graneheim & Lundman [27] was used to carry out the secondary analysis of the interview data. This form of analysis is especially appropriate when, as in this study, researchers are interested in systematically describing only the topics of interest indicated by the research questions, and not in obtaining a holistic overview of all of the data [28]. The data analysis was carried out in three stages using NVivo 12 for Mac (QSR International Pty Ltd, Victoria, Australia). In the first stage, the interview transcripts were read thoroughly in order to extract the sections of text in which the participants raised criticisms of the mixed methods field. These sections of text constituted the unit of analysis. In the second stage, the extracted sections were divided into meaning units, which were subsequently condensed, abstracted and labelled with codes. Each code included a description of the meaning of the code, an indicator to identify its presence in the data, and an example of a passage coded as belonging to that code. In the third stage, the codes were compared for similarities and differences, and clustered into several categories. The underlying meaning of these categories was

**Table 1. Characteristics of participants by discipline (N = 42).**

| | Education (*n* = 11) | Nursing (*n* = 10) | Psychology (*n* = 11) | Sociology (*n* = 10) | Total |
|---|---|---|---|---|---|
| Sex, *n* | | | | | |
| Male | 5 | 2 | 6 | 7 | 20 |
| Female | 6 | 8 | 5 | 3 | 22 |
| Geographic location, *n* | | | | | |
| North America | 8 | 4 | 7 | 3 | 22 |
| Europe | 3 | 4 | 4 | 7 | 18 |
| Oceania | 0 | 2 | 0 | 0 | 2 |
| Position, *n* | | | | | |
| Professor | 8 | 5 | 5 | 8 | 26 |
| Associate Professor | 1 | 2 | 2 | 1 | 6 |
| Assistant Professor | 1 | 1 | 0 | 1 | 3 |
| Other | 1 | 2 | 4 | 0 | 7 |
| Years since PhD graduation, *n* | | | | | |
| Less than 15 years | 3 | 8 | 5 | 3 | 19 |
| 15 years or more | 8 | 1 | 5 | 5 | 19 |
| Does not have a PhD | 0 | 1 | 1 | 2 | 4 |
| Methodological expertise, *n* | | | | | |
| Quantitative | 2 | 1 | 3 | 2 | 8 |
| Qualitative | 7 | 1 | 2 | 3 | 13 |
| Equal quantitative and qualitative | 2 | 8 | 6 | 5 | 21 |
| Experience in carrying out mixed methods studies | | | | | |
| Yes | 11 | 10 | 9 | 9 | 39 |
| No | 0 | 0 | 2 | 1 | 3 |

then examined and formulated into themes. These themes represented the study participants' criticisms of mixed methods as a field, which were the focus of RQ1. Decisions made in this phase of the study, along with any disagreements, were discussed by the researchers until a consensus was reached.

In order to answer RQ2, a multiple correspondence analysis [29–31] was carried out. This technique is a non-inferential form of statistical analysis designed to analyze the multivariate association of categorical variables by generating a representation of the underlying structure of a dataset. Since the statistical requirements of multiple correspondence analysis (i.e., sampling, linearity, and normality) are highly flexible, this method is especially suited for examining qualitative interview data transformed into quantitative data [32]. The output of the multiple correspondence analysis is a scatterplot representing the spatial grouping of categories and participants. The distances between plotted points represent the degree of similarity in the patterns of participants' responses. Multiple correspondence analysis was used to examine the relationship between the participants' discipline and the themes relating to criticisms. To perform this analysis, we used the NVivo "matrix coding query" function to generate a matrix in which binary codes related to criticisms identified in the qualitative content analysis (the mention or failure to mention the criticism) were displayed in the columns, while the 42 participants were displayed in the rows. The matrix output was exported to XLSTAT Version 2018.1 (Addinsoft, Paris, France), which was used to perform the analysis, using the binary codes for the criticisms as active variables and the participants' discipline as supplementary variables. Following the recommendations of Bazeley [32], after carrying out the multiple

correspondence analysis, we checked the results against the qualitative data to verify the interpretation of the statistical analysis.

# Findings

## RQ1; What criticisms of the mixed methods field are made by researchers in education, nursing, psychology, and sociology?

Eleven criticisms of how some members of the mixed methods community have developed and conceptualized this research approach were identified in 27 of the 42 interviews included in this secondary analysis. These criticisms were then grouped into the four domains used by Creswell [33] to map the landscape of mixed methods research: (1) the essence of mixed methods research (definitions and nomenclature), (2) philosophy (philosophical assumptions and paradigmatic stances), (3) procedures (methods and techniques for carrying out mixed methods research), and (4) politics (justification of the use of mixed methods research). Each of these four domains and the corresponding criticisms are discussed in the following sections with the support of verbatim quotes from the interviews. Table 2 shows the criticisms for each domain and the number of participants making each criticism.

**Domain 1: The essence of mixed methods research.** *Criticism 1*: *The accepted definition of mixed methods research takes into account only the mixing of both quantitative and qualitative methods*. Some participants objected that the most common definition of mixed methods research that usually prevails in the literature conceives the approach as being limited to the use of quantitative and qualitative methods. These participants believe that the field should adopt a broader definition that would also include the mixing of methods within the same tradition in a single design, that is, the combination of two or more quantitative methods or qualitative methods:

> "(. . .) for me mixed methods is not only mixing qualitative and quantitative methods, but it could also be qualitative + qualitative or quantitative + quantitative methods" (Sociologist #4).

These participants argued that the current definition of mixed methods implies that mixing the two distinct families of methods is often the only appropriate approach while in fact this

**Table 2. Types of criticisms of the mixed methods field raised by the participants.**

| Domain | Criticism |
|---|---|
| 1. The essence of mixed methods research (n = 11) | 1. The accepted definition of mixed methods research takes into account only the mixing of both quantitative and qualitative methods (n = 5) |
| | 2. The terminology used in mixed methods reflects a lack of agreement among its proponents (n = 5) |
| | 3. Mixed methods research is not a new type of methods practice (n = 2) |
| 2. Philosophy (n = 14) | 4. Mixed methods research is not a third paradigm (n = 13) |
| | 5. Current discussions of mixed methods research conceive quantitative and qualitative research as separate paradigms (n = 7) |
| | 6. Superficiality of pragmatism (n = 3) |
| | 7. Mixed methods research aligns with positivism (n = 2) |
| 3. Procedures (n = 13) | 8. Limitations of typologies (n = 8) |
| | 9. Procedures described in the literature are not aligned with mixed methods practice (n = 7) |
| 4. Politics (n = 11) | 10. Mixed methods research is not better than monomethod research (n = 8) |
| | 11. Homogenization of mixed methods research (n = 3) |

definition obviates the contingent nature of research. Certain research questions might be better answered by using a combination of methods from a single tradition. Furthermore, combining two methods from the same tradition can be as valuable and as challenging as combining two methods from different traditions. One participant used the term "pressure" to describe the feeling that he was obliged to mix quantitative and qualitative methods even when this approach was not the most appropriate one:

*". . .combining methods isn't just a matter of combining quantitative and qualitative methods. You can combine different methods that are both qualitative or both quantitative and that's, that's valuable in it- itself, and I am worried about the kind of pressure to combine quantitative and qualitative as if that would always be appropriate"* (Sociologist #2).

*Criticism 2*: *The terminology used in mixed methods reflects a lack of agreement among its proponents*. Several participants noted the lack of clear agreement on the terminology generally used to describe the concepts and procedures that pertain to mixed methods research. They also cited a tendency to use multiple definitions for the same term and different terms to refer to similar notions. One participant cited as problematic the use of several different terms (e.g., legitimation, validity, rigor) to refer to the quality of mixed methods research:

*"I would like to see a word that's used by as many people as possible to describe that [quality]. . .. But, you know, I, I just think if we, everybody continues to use different terms, that could be problematic"* (Educationalist #8).

According to this participant, while synonymous terms might add some precision when used to describe the complexities of implementing mixed methods research, their use can also generate confusion, especially among reviewers, editors and researchers who are trying to familiarize themselves with the field:

*". . .it just gets to the point where if everyone has a different definition, then how useful is that? And that gets confusing for those who review manuscripts, or editors, when people are using in different ways that exact same term"* (Educationalist #8).

Participants suggested two possible reasons for this lack of agreement. First, the tendency among some scholars to consider that mixed methods researchers should be able to use whatever terminology they may find convenient. Second, the desire of some authors to claim priority for the terminology that defines a particular method or typology. In order to resolve this lack of agreement, participants suggested that members of the mixed methods community should work towards building a greater consensus on terminology:

*"There needs to be a common language"* (Educationalist #9).

*Criticism 3*: *Mixed methods research is not a new type of methods practice*. Some participants noted the tendency in the literature to present mixed methods as a new type of research practice that emerged during the past three decades. They pointed out that the use of mixed methods has a prior history that considerably predates the time when it became formalized as a research field. These participants cited examples of studies in sociology by Jahoda and Zeisel (Marienthal study of unemployment) and fieldwork in anthropology by Margaret Mead, both dating from the early 20th century. While these studies had an influence on methodology in the social sciences on account of the ways in which they creatively combined multiple

quantitative and qualitative data sources, they have been generally overlooked in the mixed methods literature:

> *"I don't particularly think that [mixed methods research has allowed us to answer research questions which were left unanswered in the past] but what I do think is that, you know, do remember as well that mixed methods research does actually have a long history in Sociology"* (Sociologist #7).

**Domain 2: Philosophy.**    *Criticism 4*: *Mixed methods research is not a third paradigm*. A considerable number of participants argued against the idea of characterizing mixed methods research as a third paradigm. They found two major faults with this characterization. First, it relies on the idea of mixed methods research as an approach that is distinct from quantitative and qualitative methodologies. In the view of these participants, mixed methods approaches do not rely on singular elements that are distinct in their nature, philosophy, or procedures:

> *"So no, I think, ultimately, I'm probably, I'm not really convinced that is a distinct methodology (. . .) So I worry when, when the idea of something that's very special about mixed methods is given a lot, is given too much primacy"* (Sociologist #3).

> *"I don't think it's helpful to see it as a separate approach in terms of actually conducting, you know, planning and conducting, the research. . .I certainly think it's stretching it to see it as a different, as a separate paradigm. . .I think the whole idea of 'paradigm' is a little bit difficult"* (Sociologist #5).

Second, the conceptualization of mixed methods as a paradigm presupposes a strong link between epistemology and method, that is, the identification of the use of mixed methods with a particular epistemological or ontological view, whereas, in fact, these are separate entities. Attaching epistemological and ontological assumptions to mixed methods research would weaken its functionality and creative potential:

> *". . .if we restrict mixed methods to only one paradigm then we're bottlenecking mixed methods into a certain area, and we restrict the functionality of it"* (Nurse, #10).

*Criticism 5*: *Current discussions of mixed methods research conceive quantitative and qualitative research as separate paradigms*. Related to the previous criticism, a number of participants noted that current conceptualizations of mixed methods take for granted the nature of the quantitative and qualitative approaches, conceiving them as separate paradigms based on particular philosophical assumptions, thus reinforcing the conventional divide between them and accentuating their differences:

> *". . .the whole purpose, of course, of mixed methods is that it's, that's a paradigm, but I'm not convinced it is because it still draws on those conventional traditional paradigms. . .I find that's likely less helpful because again it starts from the assumption that there is a strong division between qualitative and quantitative research"* (Educationalist #2).

These participants stated that the mixed methods literature may have uncritically incorporated the methodological "rules" (conventions) that were dominant in the 1980s by associating qualitative research with the constructivist paradigm and quantitative research with the positivist paradigm. This linkage between philosophy and method may have been a result of the process of formalization of the methodology carried out by the "second generation of mixed

methods researchers" (from 1980s to present), while the "first generation" (i.e., from the 1900s to 1980s) might not have had a philosophical problem:

*"I think the biggest problem that mixed methods research is in right now is having adopted, without reflection the rules that were established in the mid 80's on, on paradigms in quali and quanti. . .we have these pillars, these quali-quanti pillars and we're working on these rooms. . .All these classical studies [from the first generation] had no problems in doing quali-quanti, it was only the attempt to formalize it which has actually created these, these problems"* (Sociologist #6).

According to this participant, the association of the quantitative and qualitative approaches with particular epistemological stances contradicts the very nature of the mixed methods approach: if such philosophical and methodological differences between quantitative and qualitative research really existed, then the integration of the two approaches would not be possible:

*". . .the big problem with having adopted this [association], on the one hand, it actually makes mixed methods impossible. So, it is not possible within one single design to argue that your da-, that there is a single and objectifiable reality out there, on the other hand, and there are multiple or no reality, there's no reality"* (Sociologist #6).

In the view of another participant, part of the mixed methods literature may have accentuated the differences between the two methodologies by representing their characteristics in different columns in a table, while ignoring the existence of methods that incorporate features of both approaches (e.g., survey containing both open and closed ended questions or qualitative studies that include descriptive statistics):

*"I know what's been recently suggested in the literature (. . .) I'm not even sure that I would say that we should have drawn a line between qualitative and quantitative as firmly as we have. A lot of the qualitative work that I do includes descriptive statistics"* (Educationalist #7).

*Criticism 6: Superficiality of pragmatism.* Some participants argued that authors in the mixed methods community sometimes characterize the notion of pragmatism in a superficial way by reducing it to merely eclecticism and confusing it with "practicalism". In this way, these authors advocate a "what works" approach which may be useful when justifying the integration of the quantitative and qualitative methods, but this attitude distorts the nature of pragmatism by failing to consider its underlying theoretical and philosophical assumptions:

*". . .they tend to think that pragmatism is just the practicalities, and it's just the technicalities. . ."* (Educationalist #2).

One participant noted that the feebly argued debates on pragmatism to date may have led the mixed methods community to undervalue the important contribution this paradigm has made to the philosophical basis of empirical inquiry:

*"I've probably never in my life seen such weak debates on pragmatism as I have in mix-, in the mixed methods debate. I mean if I think of this fabulous contribution that, that pragmatism as a philosophical discipline has made"* (Sociologist #6).

Moreover, another participant observed that many researchers in the field have acquired their knowledge of pragmatism mainly from the descriptions of the mixed methods paradigm found in the literature, whereas a sound basis for pragmatism in mixed methods research practice would require consulting the seminal papers on pragmatism, such as those by John Dewey, Charles Sanders Pierce or William James:

*"... from what I've read anything about pragmatism that's in a mixed methods paper does tend to be superficial...you have to go right back to the original authors of pragmatism and I think sometimes when we speak about pragmatism in mixed methods research, students particularly ten- tend, maybe just read some articles in pragmatism and think they know about this, but I think it is important to go right back to Dewey and James and Pierce"* (Nurse, #6).

*Criticism 7*: *Mixed methods research aligns with positivism*. A few participants noted that some members of the mixed methods community tend to accord a higher status to the quantitative component because they consider that it is more objective and more closely embodies the scientific method. In their view, some researchers regard the qualitative component as mainly a supplement to the quantitative component. Consequently, researchers may fail to appreciate the added value that may be gained by using mixed methods research:

*"qualitative research [is often used] to almost to kind of flesh out the, the, the quantitative aspects, so it's a kind of embellishment rather than seeing it as something that might challenge some of the quantitative findings or might contribute to, to ultimately rephrasing the research question or to reanalyzing the, the quantitative data"* (Sociologist #5).

**Domain 3: Procedures.** *Criticism 8*: *Limitations of typologies*. A number of participants criticized the tendency of some authors to present mixed methods designs and procedures from a typological perspective. Typologies are used in the mixed methods literature as classifications of methodological features, such as the timing and priority of the quantitative and qualitative components and the stage at which integration is carried out [7,9]. In the view of those participants, typologies are presented in the literature in a way that is excessively mechanical and prescriptive, unnecessarily simplifying the process of carrying out a mixed methods study by suggesting that a successful implementation of a mixed methods design can be carried out only by following a predefined set of steps:

*"I'm arguing against approaches that I think are too sort of mechanical in the sense of laying out: 'Ok, here's categories A, B, C, D and E, and here are the rules for applying them. And if you just follow the rules, then you'll be ok'"* (Educationalist #1).

Participants noted that this approach entails four problems. First, in order to adapt their approach to the research questions that they need to answer, researchers may need to modify the guidelines suggested in the literature. Therefore, guidelines for the use of mixed methods designs should be only "guiding principles" that are adaptable to varying circumstances and able to take into account the interactions between the different elements of the design. One of these participants stated the following:

*"I think, I mean, I started off by using...the sort of prescription...and it's only when you start to get delve more into mixed methods... So, it's, it's really, I think [they should be] just guiding principles"* (Nurse, #5).

Second, the typologies may curtail the creativity of researchers by restricting them to a series of predefined models that are considered the "correct" ways of combining quantitative and qualitative methods. As expressed by the following participant:

*". . .researchers are using mixed methods in such creative ways, it's like, it's just, when you read these designs and they can be just so, so different and they just don't fit into, you know, the typologies"* (Nurse, #3).

Third, rather than being empirically generated by examining how mixed methods research is actually carried out in practice, these typologies are the highly formalized result of a list of ideal designs formulated by mixed methods theorists, as noted by this participant:

*"There were basically two different approaches [to the development of typologies] and the one that was most common was the sort of develop very formal systems. . .The opposite of that was Bryman who went out and interviewed qualitative researchers about what they did. . .he talked to people about what they really did rather than coming up with formal systems"* (Sociologist #1).

Fourth, the existing typologies are too extensive, which makes them difficult for inexperienced researchers to apply, as we can see in the following quote:

*". . .there must be forty or fifty different designs associated with mixed methods and I think it's, you know, I think that's confusing to people and it's. . .in some way I think it becomes irrelevant"* (Psychologist #9).

*Criticism 9: Procedures described in the literature are not aligned with mixed methods practice.* Some participants mentioned occasional discrepancies between the procedures explained in textbooks and articles and the implementation and reporting of mixed methods in practice, which may not always conform to published guidelines and typologies.

*"you, you open any textbook. . .and the rules that are proposed there are broken every day very successfully by researchers who, who actually conduct the research. . .the practice and the debate need to run parallel and they probably, right now I think they are a bit too separate from each other"* (Sociologist #3).

Participants attributed this disjuncture to the fact that a few influential authors probably lacked sufficient practical experience in using mixed methods. These authors may have tended to suggest methodological guidelines "from their desk" without testing them in practice or reviewing the empirical work of other researchers:

*". . .there might be a gap that, that a lot of researchers talk about using mixed methods, but I'm not sure if they actually do it in practice"* (Psychologist #2).

Furthermore, participants also noted that in some cases those authors may have placed greater emphasis on the philosophical and theoretical basis of mixed methods than on describing the techniques involved in implementing mixed methods research:

*". . .people spend far too much time talking about epistemology, most of those discussions are actually very simple, but people make them very complicate. . .I think. . .technical questions*

*about how you work with the data and what it means [are more important]"* (Educationalist #6).

**Domain 4: Politics.** *Criticism 10*: *Mixed methods research is not better than monomethod research*. Some participants pointed out a tendency among some members of the mixed methods community to consider this type of research to be inherently superior to monomethod research:

*". . .thinking about the, the papers that I've reviewed have been for the kind of applied end journals. . .I think the main issue for me has been in terms of, you know, the purpose of using mixed methods; that I think there's a tendency to slip into thinking that more is necessarily better"* (Sociologist #9).

Participants noted that to attribute a higher status to mixed methods research is wrong because this view could lead to the oversimplification of other approaches, which would undermine their prestige. Furthermore, participants argued that a mixed methods approach is not always the best research option and a fully integrated design may not be the most appropriate. What really determines the suitability of an approach or a design is the research question of a study, so that a monomethod design is sometimes the most appropriate.

*Criticism 11*: *Homogenization of mixed methods research*. A few participants criticized a tendency in the mixed methods field to homogenize terminology and procedures. In their view, some members of the community have tried to develop a "mixed methods way of doing things" which would be acceptable to all researchers and would require them to write in a particular way using particular terminologies and strategies:

*"They're, they're trying to develop a language, they're trying to develop an approach, a strategy that, that is going to be acceptable by all mixed methods researchers, which really is, is unacceptable"* (Educationalist #2).

This attitude towards homogenization of mixed methods research could hinder the advancement of the field since it promotes a uniform approach, suppresses intellectual disputes and ignores the diversity of approaches and attitudes regarding mixed methods found in the literature. As one participant argued, to find space for legitimate difference in the field is very difficult due to the protectionist attitude of some prominent authors who are interested in propagating their own ideas rather than incorporating the ideas of other authors:

*"I fear that there's, among, among those who have some prominence, there are some who would be very eager to protect their own turf and not wanting to come together for some kind of joint effort"* (Educationalist #3).

## RQ2: What differences and similarities can be identified in the criticisms reported by researchers working in different disciplines?

Of the 27 participants who raised criticisms, ten were sociologists, eight were educationalists, five were nurses, and four were psychologists. Multiple correspondence analysis was used to analyze the differences, depending on their discipline, in the types of criticisms the participants raised. Fig 1 shows the multiple correspondence analysis map for the first two axes. Highly associated categories are plotted near to one another on the basis of their loading to the corresponding axes, while the least associated categories are plotted far from one another.

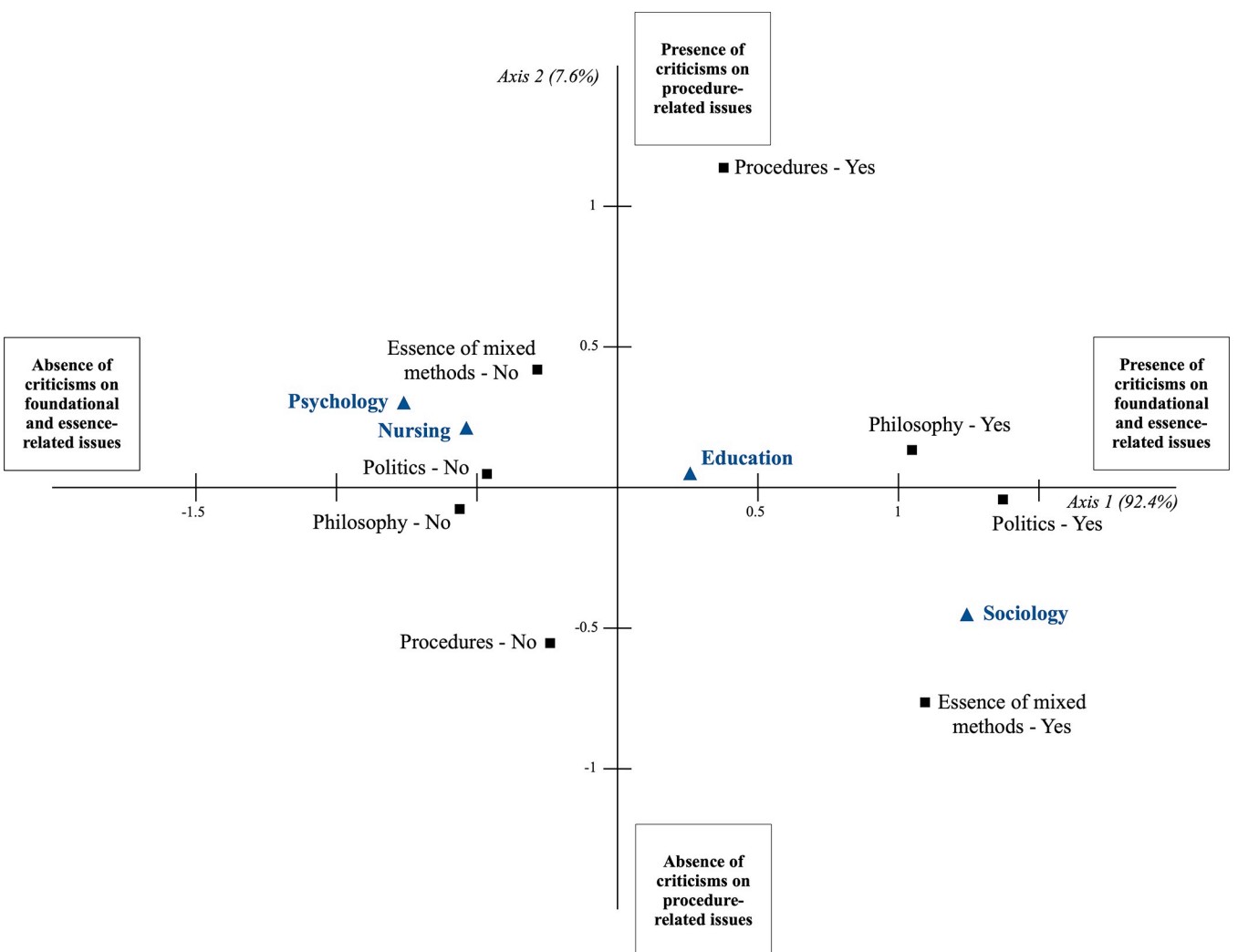

**Fig 1. Multiple correspondence analysis map of the criticisms of the mixed methods field by study participants: Active categories (black squares), discipline as supplementary variable (blue triangles).** Note: The *Yes* label indicates that the criticism was mentioned, while the *No* label indicates that the criticism was not mentioned.

Therefore, the axes should be interpreted based on the grouping seen on the map of the relative positions of the categories, which are expressed by the magnitude of the coordinates. The measures known as eigenvalues indicate how much of the categorical information is explained by each dimension. Higher eigenvalues indicate a greater amount of variance of the variables in that dimension [29,30].

On Axis 1, which accounts for most of the variation in the data (eigenvalue of 92.4%), the *yes* categories of the criticisms associated with the domains of philosophy, politics and the essence of mixed methods are plotted on the right-hand side of the map, while the *no* categories of the same domains are on the left-hand side of the map. As shown in the figure, sociologists were more likely to formulate criticisms associated with the three abovementioned domains while psychologists and nurses were less likely to formulate criticisms associated with any of these three domains. Educationalists were less inclined than sociologists to formulate criticisms associated with those domains, as indicated by the proximity of the education category to the centroid (the center of the axis). Finally, the fact that the *yes* and *no* categories of

the procedures domain are plotted on Axis 2, which has very low explanatory power (eigenvalue of 7.6%), reveals that participants in the four disciplines showed no relevant differences in formulating this criticism.

## Discussion

### Summary of findings

The aim of this study, based on a secondary analysis of interview data, was to describe the criticisms of mixed methods as a field raised by a sample of researchers in the disciplines of education, nursing, psychology, and sociology. Overall, the findings revealed a number of criticisms related to several different issues. These criticisms were initially identified by the first author in the data analysis phase of the original study, which was focused on the conceptualization and operationalization of quality in mixed methods research. The criticisms were particularly relevant since they were unprompted, that is, they were spontaneously given by the participants in response to questions about another subject, rather than to questions about criticisms. Furthermore, the original study was based on a broad and diverse sample of participants; it included a few researchers from the disciplines of sociology and education known for their criticisms as well as a considerable number of researchers whose publications seemed to be neutral on the subject of criticisms of the mixed methods field. To this latter group of researchers, the interviews afforded an opportunity to express their disagreement with some predominant notions in the mixed methods field which they otherwise might not have published. Another key element of this study is the examination of the relationship between the discipline of the participants and the frequency and type of criticisms they made. This has been possible since the sample is relatively balanced in terms of the number of participants from each of the four disciplines included in the study.

In response to RQ1, participants raised a total of eleven unprompted critical remarks, categorized in the following four domains: the essence of mixed methods (three criticisms), philosophy (four criticisms), procedures (two criticisms), and politics (two criticisms). Nine of the eleven critical comments have been previously mentioned in the literature. For instance, on the essence of mixed methods domain, some authors have alluded to problems such as the narrow definition of mixed methods research [15,34–36], the lack of agreement on the terminology used [16,37] and the inappropriateness of considering mixed methods a new methodology [35,38,39]. In the philosophical domain, several authors have criticized the dominance of a positivist approach to mixed methods research in some disciplines [37,40,41] while some authors have pointed out that considering as a separate or distinct paradigm can lead to an artificial separation of the quantitative and qualitative approaches [42–45]. In the procedures domain, a number of authors cited the problems inherent in conceptualizing mixed methods designs typologically, since such a view is restrictive and unable to reflect the variety of mixed methods designs used in practice [38,46,47]. Finally, in the politics domain, a few authors have criticized a tendency, in some of the literature, to homogenize the field [48], while others have critically noted the occasional adoption of a universalist position based on the idea that the mixed methods approach is inherently superior to monomethod research [44,47]. However, we also identified two criticisms not previously mentioned in the literature: the excessively superficial characterizations of pragmatism (criticism 6) found occasionally in the mixed methods literature and the description of procedures that are not necessarily in line with research practice (criticism 9).

Regarding criticism 6, some participants noted a tendency on the part of some researchers in the field to cite, when writing about pragmatism, what other mixed methods researchers had written about this paradigm rather than citing foundational writings, such as those by

John Dewey, William James, or Charles Sanders Peirce. In fact, none of the most influential and most frequently cited textbooks on mixed methods research in the four disciplines we studied cites any work by key authors in the pragmatist tradition. Therefore, it could be useful to learn whether these authors' highly synthetic explanation of foundational knowledge leads inexperienced researchers to only weakly engage with this paradigm, or, on the other hand, whether this simplification might help them to grasp the basic principles of pragmatism more quickly while leading them to consult first-hand the foundational writings.

With respect to criticism 9, participants reported that researchers do not always follow in practice all of the procedures described in mixed methods textbooks. This disjuncture between textbook guidance and research practice has been described in several methodological reviews of the use of mixed methods in the four disciplines we included in our study. Features such as explicitly stating the mixed methods design used, reporting mixed methods research questions, or explicitly stating the limitations associated with the use of a mixed methods design are regarded by some authors as key characteristics of mixed methods studies (Creswell and Plano Clark, 2018; Plano Clark and Ivankova, 2016; Onwuegbuzie and Corrigan, 2014; O'Cathain et al, 2008). However, Bartholomew & Lockard (2018) reported that very few of the studies included in their review of the use of mixed methods in psychotherapy explicitly stated the mixed methods design used (13%) or reported mixed methods research questions (29%). Additionally, Bressan et al. (2017) and Irvine et al. (2021), in their reviews on mixed methods in nursing, found that most of the studies they included failed to report the limitations associated with the use of a mixed methods design. Therefore, it could be of great interest to study whether the omission of these characteristic features of mixed methods studies reflects the researchers' view that these features are unimportant, or whether they are unfamiliar with reporting standards. Identifying this latter criticism is a particularly relevant finding of this study, since the intimate context of the interview might have led to the expression of subjective judgments that otherwise might not have come to light (i.e., the participant's perception that some authors may not habitually carry out empirical research).

With respect to RQ2, we found relevant differences in the type of criticisms raised across disciplines. In fact, one of our key findings is that criticisms in the procedures domain were equally distributed across the four disciplines, while criticisms in the essence of mixed methods, philosophy and politics domains were clearly more common in sociology. First, these findings are consistent with statements made by Plano Clark and Ivankova [3] regarding the ways in which the sociocultural context of researchers — including the discipline in which they work — can shape their beliefs, knowledge and even experiences with regards to mixed methods. Indeed, the greater number of criticisms made by sociologists categorized in three of the four domains shows how disciplinary conventions might affect how researchers think about mixed methods and judge the acceptability of certain predominant conceptualizations. According to several authors [49–51], critique is a foundational and distinct feature of the discipline of sociology. Therefore, the generalized tendency among sociologists to question traditional assumptions about the order of the world and to detach themselves from predominant belief systems and ideologies might help to explain why many sociologists in our sample criticized ideas such as the conceptualization of quantitative and qualitative research as separate entities and the consideration of mixed methods as inherently better than monomethod research.

### Theoretical implications of the study

The findings of our study highlight several differences in opinion in the mixed methods field previously identified by authors such as Greene [52], Tashakkori and Teddlie [17], Leech [53]

and Maxwell, Chmiel, and Rogers [54], among others. Those authors showed that, owing to differences in philosophical and theoretical stances and their adherence to different research cultures, researchers in this field sometimes disagree on foundational issues such as nomenclature, the need for consensus, and the definition of mixed methods research. In our study, the participants, particularly those in the field of sociology, made several criticisms about how some foundational and philosophical aspects of mixed methods research have been conceptualized by the mixed methods community, including how mixed methods has been defined and accorded status as a third paradigm. Furthermore, our findings showed contradictory criticisms formulated by the participants as a group: while some researchers criticized the lack of a consensus in the field on the terminology used to describe mixed methods research (criticism 2), others criticized a tendency by some authors to homogenize terminology (criticism 11).

This divergence of views is consonant with the notion of *communities of practice* suggested by Denscombe [12]. Departing from Kuhn's notion of paradigms as "shared beliefs among the members of a specialty area" (as cited by Morgan [10]), in Denscombe's view, the broader mixed methods community is a paradigm encompassing a conglomerate of multiple research communities shaped by the principles, values and interests prevailing in their disciplines and research orientations. In line with that author's view that methodological decisions and viewpoints "will be shaped by a socialization process involving the influence of peers" [12], our findings suggest that the disciplinary community of our participants is likely to have informed their criticisms of the mixed methods field. Although this divergence of views suggests that complete agreement and unhindered communication among researchers is not possible [55], Ghiara [56] argued that, in Kuhn's view, some form of communication is always possible; and furthermore, conflicting viewpoints can be reconciled to a certain extent. Similarly, Johnson [57] has argued that this diversity of critical voices, rather than being a problem, merely indicates that "reality is likely plural"—there is no single set of ontological assumptions underlying mixed methods—and, furthermore, knowledge is articulated on the basis of "multiple standpoints and strategies for learning about our world" that can be reconciled to some degree. An example of the healthy coexistence of divergent viewpoints within the mixed methods community can be found in the *Journal of Mixed Methods Research*, the leading journal in the field, described in its webpage as a "primary forum for the growing community of international and multidisciplinary scholars of mixed methods research". The journal publishes a wide range of manuscripts, including articles revealing approaches to mixed methods research that rest on divergent foundational and philosophical perspectives.

The desire for inclusion of divergent viewpoints should not lead researchers to ignore the challenges posed by this divergence. Due to the interdisciplinary nature of mixed methods research, Curry et al. [58] argue that mixed methods teams often include researchers with different methodological backgrounds and propensities. Occasionally, these differences may pose challenges for establishing effective collaboration and for efficiently integrating research methods. A greater awareness of multiple perspectives on mixed methods research, including divergent critical views like those reported in our study, could help researchers better anticipate difficulties that might present themselves in the course of working with researchers who hold differing viewpoints. Furthermore, as Maxwell, Chmiel, and Rogers [54] have suggested, a better understanding on the part of mixed methods researchers of the perspectives of others in the field who embrace a differing approach should facilitate the process of integrating quantitative and qualitative methods in studies where different ontological positions coincide. In a similar vein, such an understanding could also help overcome a form of *methodological tokenism* described by Hancock, Sykes and Verma [59]. This can occur when mixed methods researchers fail to attend to, and therefore align, the distinct epistemological and ontological

premises that underlie the methodological orientations that are integrated in a mixed methods design.

Furthermore, the recognition of mixed methods researchers' divergence of views should be an integral part in any effort to design and implement a curriculum for mixed methods research. Plano Clark and Ivankova [13] have pointed out that any lack of clarity concerning the existing disagreements about foundational elements of the mixed methods approach could be confusing to researchers inexperienced in this field. Therefore, it is essential that courses and workshops on mixed methods research take note of these criticisms. This last point is particularly important since the topic of critical viewpoints is not included in any of the mixed methods syllabus exemplars published in the literature, including those by Earley [60], Christ [61], and Ivankova and Plano Clark [13].

## Limitations, strengths, and possibilities for future research

Our findings are subject to a few limitations. First of all, the interviews were carried out by Skype and telephone. While these two forms of communication allowed us to interview participants residing in various locations around the world, they limited the possibilities for building the sort of rapport that might have encouraged some participants to elaborate more in their responses. To minimize this limitation, participants were given the opportunity in the member-checking phase to add additional insights to their initial statements. Second, the transferability of our findings is limited by our decision to include only four disciplines while excluding the views of researchers working in other disciplines that also have a high prevalence of mixed methods studies, such as medicine, business, and information science. From our findings alone, it is not possible to infer how frequent these criticisms are and what types of criticisms may be more prevalent in each of the disciplines. Third, multiple correspondence analysis is an exploratory method not appropriate for testing hypotheses or statistical significance. In other words, the method is designed to describe associations between categorical variables rather than to make predictions about a population [31]. Therefore, in light of the limitations of multiple correspondence analysis for drawing deeper inferences, the findings regarding RQ2 should be considered provisional and subject to further investigation. Finally, since this study is based on a secondary analysis, the interview questions did not specifically prompt the participants to bring up criticisms since the questions were focused on participants' views related to the quality of mixed methods research. If we had specifically prompted participants to report their own criticisms, it is likely that more critical opinions would have been gathered.

While the use of secondary data entailed certain limitations, it also conferred some advantages. Participants' critical statements were entirely spontaneous since they were not explicitly solicited. This spontaneity probably helped to reduce the social desirability bias, that is, a presumed tendency for respondents to dissimulate their own critical views in a way that might seem professionally and socially more acceptable. Furthermore, despite the limitations of multiple correspondence analysis, this method allowed us to generate a parsimonious visual representation of the underlying patterns of relationships among the criticisms and the disciplines. This representation helped us to improve our interpretation of the qualitative findings and to identify useful leads for carrying our further analysis. A further strength is that the study included a broad sample of participants in terms of their geographic location, academic position, seniority, and methodological expertise. This diversity probably also afforded us access to a wider range of views. Finally, a key strength of our study is that, to our knowledge, it is the first empirical study that has addressed the topic of criticisms of mixed methods as a field.

This study represents a step towards a better understanding of some current criticisms of mixed methods research. However, further research will be needed to confirm and expand our findings. Future research based on a larger and more diverse sample of mixed methods researchers could extend the scope of our research questions and help researchers generalize from our findings. Such studies might help us discover whether researchers from other disciplines share the criticisms made by the participants in our study and whether those researchers harbor other criticisms of their own. Other analytical tools could be used to examine in greater detail how the circumstances and attributes of researchers — including their disciplinary background, methodological expertise and paradigmatic viewpoints — influence the way these scholars formulate their criticisms of the mixed methods field. Finally, future research is needed to examine the critical accounts of researchers less experienced in methodological writing and probably less exposed to current theoretical debates and developmental issues in the field.

## Acknowledgments

The authors would like to acknowledge the help of Dick Edelstein in editing the final manuscript. The first draft of this paper was written while the first author was a visiting scholar in the Mixed Methods Program of the University of Michigan, directed by Michael D. Fetters, John W. Creswell, and Timothy D. Guetterman. This author would like to thank these three scholars for their support and, particularly Michael D. Fetters for his invaluable mentorship over the past few years.

## Author Contributions

**Conceptualization:** Sergi Fàbregues, Elsa Lucia Escalante-Barrios, José Francisco Molina-Azorin, Quan Nha Hong, Joan Miquel Verd.

**Data curation:** Sergi Fàbregues.

**Formal analysis:** Sergi Fàbregues, José Francisco Molina-Azorin.

**Investigation:** Sergi Fàbregues.

**Methodology:** Sergi Fàbregues, Elsa Lucia Escalante-Barrios, José Francisco Molina-Azorin.

**Software:** Sergi Fàbregues.

**Supervision:** Joan Miquel Verd.

**Validation:** Sergi Fàbregues, Elsa Lucia Escalante-Barrios, José Francisco Molina-Azorin, Quan Nha Hong.

**Writing – original draft:** Sergi Fàbregues.

**Writing – review & editing:** Sergi Fàbregues, Elsa Lucia Escalante-Barrios, José Francisco Molina-Azorin, Quan Nha Hong, Joan Miquel Verd.

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
