## [Decision Letter · Decision Letter 0]

13 Nov 2020

PONE-D-20-22907

Taking a critical stance towards mixed methods research: A cross-disciplinary qualitative secondary analysis of researchers’ views

PLOS ONE

Dear Dr. Fàbregues,

Thank you for submitting your manuscript to PLOS ONE. After careful consideration, we feel that it has merit but does not fully meet PLOS ONE’s publication criteria as it currently stands. Therefore, we invite you to submit a revised version of the manuscript that addresses the points raised during the review process.

Your manuscript was reviewed by three accomplished scientists with mixed methods expertise. I also reviewed the paper and I concur with the vast majority of all comments, critiques and suggestions raised by the reviewers. This is a highly important manuscript that presents much needed findings in an area seldom studied. In addition to the reviewer comments, I have a major concern about the multiple correspondence analysis. How reproducible are the MCA results? I would suggest that the authors more explicitly acknowledge that this part of the paper is more exploratory than thematic/content analysis. Lastly, I strongly agree with Reviewer 1's concern about use of the term pedagogy and the associated text in the discussion. The dissimilarity between practice and textbook descriptions of mixed methods is notable, but the authors should be more precise with use of language here. 

Thank you for your time and dedication in preparing this important work. We look forward to receiving your revised manuscript.

Kind regards,

Adam T. Perzynski, PhD

Academic Editor

PLOS ONE

Journal Requirements:

Reviewers' comments:

Reviewer's Responses to Questions

**Comments to the Author**

1. Is the manuscript technically sound, and do the data support the conclusions?

Reviewer #1: Partly

Reviewer #2: Yes

Reviewer #3: Partly

2. Has the statistical analysis been performed appropriately and rigorously? 

Reviewer #1: Yes

Reviewer #2: I Don't Know

Reviewer #3: Yes

3. Have the authors made all data underlying the findings in their manuscript fully available?

Reviewer #1: Yes

Reviewer #2: No

Reviewer #3: Yes

4. Is the manuscript presented in an intelligible fashion and written in standard English?

Reviewer #1: Yes

Reviewer #2: Yes

Reviewer #3: Yes

5. Review Comments to the Author

Reviewer #1: Please see my full review uploaded as attachment. In summary, this is an interesting study that provides a more transparent window into some of the recurring debates within and among mixed methods researchers and their critics and detractors, but perhaps more importantly, interview data reveals aspects of mixed methods critiques that have not been as explicitly or formally stated in the published research literature, and the findings raise some important operational, conceptual, professional, and practical implications. However, the credibility and validity of the work presented in this manuscript is hampered by the authors’ editorializing recommendations and, at times, apparently unfounded interpretations of the empirical data. The mainly descriptive understanding gained from the study itself warrants dissemination without the need to overstep the bounds of what these empirics allow the authors to infer. The study authors describe was systematic, methodologically sound, and proceeded from a careful research design that affords reasonable confidence in the validity of descriptive results and makes the limits of generalizability and potential sources of bias clear. Yet, subjective interpretations beyond the data amount to a form of advocacy that the study at hand simply does not warrant.

Based on my review of the work in its present form, I recommend that PLOS ONE request that authors Revise & Resubmit, with Major Revisions, Manuscript Number: PONE-D-20-22907, in order to address this main weakness, with a focus on rethinking and rearticulating the nature of the implications of the findings in the current Discussion section (pp.23-29) and Abstract.

Reviewer #2: In this study, the authors reported the result of an analysis of the criticisms of mixed methods research using a cross-national purposive sample of 42 researchers from multiple disciplines (education, nursing, psychology, sociology) who have published on the methods of mixed-methods research. The researchers originally participated in an interview regarding quality in mixed-methods research and were then recontacted and, if they agreed, provided additional information to their original responses. Eleven different responses were identified by the authors and then grouped into 4 domains used by an expert in mixed methods research (Creswell) to map the mixed methods research landscape: essence, philosophy, procedures, and politics. Criticisms related to essence, philosophy, and politics were more prevalent among sociologists. Criticisms related to procedures were equally distributed across the disciplines. Based on their findings, the authors made four recommendations: address procedural criticisms, be more tolerant of the diverse ways to conceptualize mixed methods, account for other disciplines’ criticisms; and take note these mixed methods criticisms in training/educational programs.

Mixed methods have become increasingly more common and promoted in research, and the authors are to be commended for investigating criticisms of these approaches among a sample of investigators from different parts of the world and from different disciplines. This is an interesting study. My concerns mainly revolve around making the language a little more accessible to the wide readership of PLOS by explaining terminology used, and providing additional detail so that readers may better understand the methods and findings. If it is possible (in terms of word limits), the manuscript would be enhanced if the authors utilized more quotations to illustrate their findings and allay any reader concerns that many of the quotations are coming from the same individuals. My specific concerns follow:

1. Line 35: It would be helpful to readers unfamiliar with qualitative work for the authors to describe what is meant by “typologies.”

2. Lines 74-75: Related to points #7 and #21 below, if the secondary data analysis was already “covered” by the IRB, why were participants recontacted? It appears from later in the manuscript that this contact was an opportunity for authors to collect additional information from participants? Please clarify.

3. Line 86: It would be helpful to readers if the authors explained what they meant by criterion sampling. Does this mean there were inclusion criteria?

4. Line 105: This is the data collection procedure for the original study? Please clarify.

5. Lines 106-107: There seems to be some missing text, and there is no (a).

6. Line 109 (two interviews): In other words, each participant was interviewed twice? Please clarify.

7. Related to Point #21 below, the fact that the authors were recontacted and given opportunity to clarify their responses needs to be described here in the data collection. Presumably, it was this opportunity to clarify that necessitated recontacting participants (point 2 above) and that two individuals turned down? Please clarify.

8. Lines 164-165. It would be helpful to readers unfamiliar with Creswell’s publication for the authors to provide more descriptive information about each of the four domains. For example, what do “philosophy” and “politics” refer to here?

9. Criticism #2 in Table 2: I think it should be “proponents.”

10. General concern about the quotations: The quotations are very helpful to illustrate points. Is there a way the authors could identify the individuals, may like “Educationalist #1” or “Sociologist #3” to allay readers’ concerns that all the quotations from sociologists, for example, are not coming from the same sociologist.

11. Lines 257-259: the phrase “…presupposes a strong link between epistemology and method, whereas….” is unclear. Please clarify.

12. Lines 279-281: To help readers unfamiliar with the dominant rules of the 1980s, can the authors explain more about what the rules were.

13. Lines 312-315: For readers unfamiliar with these terms, the authors should explain what is meant by “pragmatism,” “eclecticism,” and “practialism.”

14. Line 343: The authors use the phrase “in some disciplines,” but from the quotation that follows it is not clear whether the participant was referring to entire disciplines or individual researchers. Please clarify.

15. Lines 393-406: It is a little concerning that the authors refer to participants in the plural but provide only one quotation. Is it possible to provide additional quotations to show more clearly to readers that multiple participants shared these views.

16. Lines 427-429: It is interesting that some participants complained about the tendency for homogenized terminologies and procedures, while other participants complained about the variation in terms and lack of agreement (lines 202-206). I might have missed it, but this might be important to include in the discussion: that the criticisms can be contradictory, which might make them difficult to address in a manner that will please everybody.

17. Lines 448-466: The authors need to help readers understand what Figure 1 is showing and what correspondence analysis mapping is doing. What does 92.4% actually refer to? (line 453). What do the values of -1.5 to 1.5 mean on the axes. It will be really helpful to readers if the authors can explain more clearly why a specific discipline is placed where it is relative to the axes’ scales.

18. Lines 486-487 “known for their criticisms”: Were the individuals known for their criticisms all from the same discipline?

19. Line 541 Limitations: The small numbers and the relatively small number of disciplines should be noted as limitations.

20. Line 541: It seems like a limitation that should be noted is that the sample was limited to researchers who have written on the methods of mixed-methods research, especially because one of the criticisms was that the people who are writing about mixed methods aren’t necessarily the ones who are really doing mixed methods research.

21. Line 547-548: The ability to provide additional insights needs to be included in the methods section. Is this what the two persons refused to do?

22. Line 567 “typologies”: Was this typologies of methods? Please clarify.

23. Maybe I missed it, but do the authors have ideas for future research in this area?

24. Figure 1: Shouldn’t the axes be labeled in some way?

Reviewer #3: Thank you for the opportunity to review this manuscript examining criticisms of the field of mixed-methods research. The manuscript is well written and clear, and is grounded in prior articulations of controversies embedded within mixed-methods research. Less clear, however, are the ways in which the authors see this study’s findings advancing the field, and the extent to which the study is sufficiently distinct, as a secondary data analysis, from the primary parent study. Additional details are as follows.

- In the introduction, the authors describe two existing summary overviews of criticism of mixed-methods research but that, to date, no empirical examination of this domain has been carried out. While the fact that prior work, until now, has discussed this topic in terms of theory and methodology is notable, I am not sure that the authors have provided a convincing rationale yet for why this particular study is useful. This seems important to explicate, especially given the nearly complete overlap of the study’s findings with these two prior syntheses.

- Could the authors clarify how they inferred the methodological expertise of the participants?

- From the authors’ stated inclusion criteria, participants from other disciplines that commonly employ mixed methods research (public health, social welfare, etc.) were excluded from the study. Though it may be unrealistic to expect a study like this to include participants from comprehensive disciplines, could the authors add some clarification on their defined focus on education, nursing, psychology, and sociology?

- I recommend that the authors add clarification and explanation of how, exactly, the current study really diverges from the parent study such that it constitutes a qualitative secondary data analysis rather than a presentation of residual findings from the parent analysis already published. The authors may wish to reference Williams and Collins (2002) for an example of structuring an analysis section with more fidelity to the approach of qualitative secondary data analysis:

Williams, C. C., & Collins, A. A. (2002). The social construction of disability in schizophrenia. Qualitative Health Research, 12(3), 297-309.

- Related to the above: at multiple points the authors highlight that the criticisms they are discussing are particularly relevant because they were given spontaneously in the original study. However, the original study appears to focus on evaluations of mixed methods research. This seems like a natural setting in which criticisms and approvals would be raised– all of which is to say, I am not sure that this is as noteworthy a characteristic of the current study as the authors seem to suggest it is. Indeed, at least two of the current study’s identified criticisms were already presented as findings in the original study.

- How transferable do the authors believe their findings are for other researchers who are mixed methodologists? (Keeping in mind that the study likely excluded researchers who work at the intersection of the inclusion criteria disciplines which may affect their views and experiences of mixed methods research.)

- I am not sure that it is clear yet how the current study provides much insight beyond what Creswell and others already have established. As the authors note in the discussion, the vast majority of the criticisms they identified have previously been documented in literature about mixed methods approaches to research. If the authors don’t identify much additional insight, I think that can be ok, as mounting evidence of the need for clarity among mixed-methods research could be seen as important for the continued growth, development, and advancement of the field. But if the authors see other novel conclusions, they should state this clearly.

- Related to the above, in the discussion section the authors dedicate the majority of space to reiterating that most of their findings have been noted in other research. There is comparatively less content discussing and exploring what their two new findings (lines 515-517) mean for the domain of mixed methods research.

- There seem to be some interesting contradictions present in the study’s findings that are not currently attended to or processed in the discussion section. For instance, the finding pertaining to needing consensus on terminology (e.g., lines 225-228) within criticism 2 seems at odds with criticism 11 in which it is described that “participants criticized a tendency in the mixed methods field to homogenize terminology.” Similarly, criticism 8 focuses on the notion that literature on mixed-methods presents typologies that “unnecessarily simplif[y] the process of carrying out a mixed methods study” and “curtail the creativity of researchers,” yet criticism 9 focuses on how researchers often do not align their mixed-methods practice with procedures documented in the literature. It may make sense that a diverse sample may have differing stances, but the discussion section would benefit from explicit attention to and interpretation of these incongruencies.

- Though the discussion section discusses some practical implications of the study’s findings, this section would benefit from additional detail and substance. For example, what do the authors recommend as requisite steps to building awareness (lines 588-591)? How, exactly, are the authors suggesting that procedural criticisms (lines 566-570) should be addressed in order to improve the practice of mixed methods research?

Minor:

- On page 7, “Questions from the background, (b) participants’ conceptualization of mixed methods research, and (c) how participants perceived the quality of the mixed methods approach in practice” is not a complete sentence.

- In Table 2, 1.2., “proponent” should be “proponents”.

- Criticisms as section headings should be phrased consistently either as the researchers’ belief or the belief they’re taking a stance against. For instance, criticism 4 is “Mixed methods research is not a third paradigm” and seems appropriately worded as a criticism; however, criticism 10 is “Mixed methods research is better than monomethod research” when the criticism itself seems to be that that mixed-methods research is not better than monomethod research.

6. PLOS authors have the option to publish the peer review history of their article (what does this mean?). If published, this will include your full peer review and any attached files.

Reviewer #1: No

Reviewer #2: No

Reviewer #3: No

---

## [Author Response · Author response to Decision Letter 0]

26 Feb 2021

The Response to Reviewers file has been attached to the manuscript.

---

## [Decision Letter · Decision Letter 1]

10 May 2021

Taking a critical stance towards mixed methods research: A cross-disciplinary qualitative secondary analysis of researchers’ views

PONE-D-20-22907R1

Dear Dr. Fàbregues,

We’re pleased to inform you that your manuscript has been judged scientifically suitable for publication and will be formally accepted for publication once it meets all outstanding technical requirements. Please take note of and make necessary changes according to the minor requests made by Reviewer 3. 

Kind regards,

Adam T. Perzynski, PhD

Academic Editor

PLOS ONE

Additional Editor Comments (optional):

Reviewers' comments:

Reviewer's Responses to Questions

**Comments to the Author**

1. If the authors have adequately addressed your comments raised in a previous round of review and you feel that this manuscript is now acceptable for publication, you may indicate that here to bypass the “Comments to the Author” section, enter your conflict of interest statement in the “Confidential to Editor” section, and submit your "Accept" recommendation.

Reviewer #1: All comments have been addressed

Reviewer #2: All comments have been addressed

Reviewer #3: All comments have been addressed

2. Is the manuscript technically sound, and do the data support the conclusions?

Reviewer #1: Yes

Reviewer #2: Yes

Reviewer #3: Yes

3. Has the statistical analysis been performed appropriately and rigorously? 

Reviewer #1: Yes

Reviewer #2: Yes

Reviewer #3: Yes

4. Have the authors made all data underlying the findings in their manuscript fully available?

Reviewer #1: Yes

Reviewer #2: No

Reviewer #3: No

5. Is the manuscript presented in an intelligible fashion and written in standard English?

Reviewer #1: Yes

Reviewer #2: Yes

Reviewer #3: Yes

6. Review Comments to the Author

Reviewer #1: (No Response)

Reviewer #2: I appreciate the authors' effort to answer my concerns. These concerns have been adequately addressed.

Reviewer #3: I appreciate the opportunity to read this edited manuscript. I believe that the authors have addressed reviewers' concerns and have provided appropriate clarification, additional detail, and expanded discussion as recommended. I see this manuscript as an important contribution to the field and look forward to seeing how other scholars interpret and build upon this work.

While I recommend acceptance, I have two very minor suggestions related to Table 1: first, that the authors replace "Gender" with "Sex", and that a "Total" column might be added for a reader's very quick reference.

7. PLOS authors have the option to publish the peer review history of their article (what does this mean?). If published, this will include your full peer review and any attached files.

Reviewer #1: No

Reviewer #2: No

Reviewer #3: No

---

## [Editor Report · Acceptance letter]

30 Jun 2021

PONE-D-20-22907R1 

Taking a critical stance towards mixed methods research: A cross-disciplinary qualitative secondary analysis of researchers’ views 

Dear Dr. Fàbregues:

I'm pleased to inform you that your manuscript has been deemed suitable for publication in PLOS ONE. Congratulations! Your manuscript is now with our production department. 

Kind regards, 

on behalf of

Dr. Adam T. Perzynski 

Academic Editor

PLOS ONE